# A Novel Handrub Tablet Loaded with Pre- and Post-Biotic Solid Lipid Nanoparticles Combining Virucidal Activity and Maintenance of the Skin Barrier and Microbiome

**DOI:** 10.3390/pharmaceutics15122793

**Published:** 2023-12-17

**Authors:** Ana Carolina Henriques Ribeiro Machado, Laís Júlio Marinheiro, Heather Ann Elizabeth Benson, Jeffrey Ernest Grice, Tereza da Silva Martins, Alexandra Lan, Patricia Santos Lopes, Newton Andreo-Filho, Vania Rodrigues Leite-Silva

**Affiliations:** 1Programa de Pós-Graduação em Medicina Translacional, Universidade Federal de São Paulo, Rua Pedro de Toledo, 720, Sao Paulo 04039-002, SP, Brazil; carolina.ribeiro@unifesp.br (A.C.H.R.M.); vania.leite@unifesp.br (V.R.L.-S.); 2Departamento de Ciências Farmacêuticas, Instituto de Ciências Ambientais, Químicas e Farmacêuticas, Universidade Federal de São Paulo, Rua São Nicolau, 210, Diadema 09913-030, SP, Brazil; l.marinheiro@unifesp.br (L.J.M.); patricia.lopes@unifesp.br (P.S.L.); 3Curtin Medical School, Curtin University, Perth, WA 6845, Australia; h.benson@curtin.edu.au; 4Frazer Institute, Faculty of Medicine, The University of Queensland, Brisbane, QLD 4102, Australia; jeff.grice@uq.edu.au; 5Laboratório de Materiais Híbridos, Departamento de Química, Instituto de Ciências Ambientais, Químicas e Farmacêuticas, Universidade Federal de São Paulo, Rua São Nicolau, 210, 2° Andar, Diadema 09913-030, SP, Brazil; tsmartins@unifesp.br; 6Shanghai Pechoin Daily Chemical Corporation, Shanghai 200060, China; alexandra.lan@pechoin.com; 7School of Pharmacy, The University of Queensland, Brisbane, QLD 4102, Australia

**Keywords:** handrub tablet, solid lipid nanoparticles, virucide, skin barrier, microbiome

## Abstract

Objective: This study aimed to develop a holobiont tablet with rapid dispersibility to provide regulation of the microbiota, virucidal activity, and skin barrier protection. Methods: A 2^3^ factorial experiment was planned to define the best formulation for the development of the base tablet, using average weight, hardness, dimensions, swelling rate, and disintegration time as parameters to be analyzed. To produce holobiont tablets, the chosen base formulation was fabricated by direct compression of prebiotics, postbiotics, and excipients. The tablets also incorporated solid lipid nanoparticles containing postbiotics that were obtained by high-pressure homogenization and freeze-drying. The in vitro virucidal activity against alpha-coronavirus particles (CCoV-VR809) was determined in VERO cell culture. In vitro analysis, using monolayer cells and human equivalent skin, was performed by rRTq-PCR to determine the expression of interleukins 1, 6, 8, and 17, aquaporin-3, involucrin, filaggrin, FoxO3, and SIRT-1. Antioxidant activity and collagen-1 synthesis were also performed in fibroblast cells. Metagenomic analysis of the skin microbiome was determined in vivo before and after application of the holobiont tablet, during one week of continuous use, and compared to the use of alcohol gel. Samples were analyzed by sequencing the V3–V4 region of the 16S rRNA gene. Results: A handrub tablet with rapid dispersibility was developed for topical use and rinse off. After being defined as safe, the virucidal activity was found to be equal to or greater than that of 70% alcohol, with a reduction in interleukins and maintenance or improvement of skin barrier gene markers, in addition to the reestablishment of the skin microbiota after use. Conclusions: The holobiont tablets were able to improve the genetic markers related to the skin barrier and also its microbiota, thereby being more favorable for use as a hand sanitizer than 70% alcohol.

## 1. Introduction

Hand disinfection is an effective way to prevent a variety of diseases and has received global attention due to the recent pandemic. The Centers for Disease Control and Prevention (CDC) recommend the use of soap with water or alcohol-based handrubs (ABHRs) containing least 60% alcohol [1]. However, these sanitizing agents can negatively affect skin barrier function, causing an increase in transepidermal water loss (TEWL) and an acute loss of surface lipids and denatured proteins of the epidermis, mainly the stratum corneum. This can propagate an inflammatory response in the skin, resulting in skin diseases [1,2]. Although it is a necessary prevention measure, this recommended hygiene practice has, at the same time, increased the prevalence of inflammatory and allergenic contact dermatitis [3]. In addition, when the skin barrier is disrupted by long-term use of sanitizers, the beneficial bacteria in the skin flora can become virulent, increasing the risk of contamination [4]. The use of hand sanitizers with the addition of barrier protectants and avoiding new allergens are therefore recommended as an alternative to common detergents [5].

A plethora of microorganisms can be found on our hands, with bacteria being the most prevalent (>80% relative abundance), then viruses, and then fungi (<5% relative abundance) [6]. The most prevalent bacteria are from the *Staphylococcaceae*, *Corynebaccteriaceae*, *Propionibacteriaceae*, and *Streptococcaceae* families [7]. Information regarding viruses is more limited as the identification of viral groups on the skin is difficult, due to the low biomass obtained from skin samples [8]. It was reported that up to 94% of the viral sequences did not match a known viral genome in reference databases [9]. It is also worth noting that these studies focused solely on the DNA virome, and the RNA virome remains uncharacterized [8]. However, it is important to note that not all microorganisms are bad. An appropriate balance between a human as the host and its microbiota, which together are described as the holobiont, contributes to health and wellness [10]. Repeated use of hand sanitizer can upset this balance; therefore, one way of promoting or reestablishing the balance of hand microbiota might be to incorporate active ingredients as prebiotics and postbiotics, carefully selected based on their ability to stimulate the growth of natural skin microbiota on human hands. 

We have considered the key elements in the design of a new approach for hand sanitizer that effectively disinfects the skin of harmful microorganisms whilst protecting the skin barrier and promoting a healthy skin microbiota for long term skin health. 

First, consideration of an effective anti-infective agent. Quaternary ammonium compounds, such as cetyltrimethylammonium chloride, are the most useful and widely accepted antiseptics [11]. They are effective against viruses by the interaction of their cationic charges with the virus lipid membrane, causing membrane disruption and protein denaturation [12].

Second, an improved delivery system for the skin. Whilst most antiseptics are conventional formulations, antimicrobial nanocarriers have been studied for their potential to provide more effective deliveries. For example, Steelandt et al. [13] evaluated a chlorhexidine-loaded polymeric nanocapsule (CHX-NC) suspension formulation that was prepared by a new solvent-free process, easily freeze-dryable, and stable over one month storage. They showed that the in vitro inhibition of two bacterial strains (*Escherichia coli* and *Staphylococcus aureus*) and one fungal strain (*Candida albicans*) was equivalent between the nanocapsule formulation and chlorhexidine solution, with the former providing advantages for biocompatibility and ease of use. Hirao et al. [12] synthesized mesoporous silica particles filled with cetyltrimethylammonium chloride, a cationic surfactant with known antiseptic activity. They showed high loading and retention of the surfactant, with antiviral activity up to 3 months and the potential to use the particles in a range of applications, including wallpaper and air filters. SLNs composed of biodegradable and safe lipidic components are also promising nanocarriers for controlled delivery of drugs and can be designed to carry a variety of hydrophilic and lipophilic substances within their internal core structures. SLNs also have the advantage of being solvent-free [13]. SLNs are composed of solid lipids, analogous to skin lipids, which are dispersed in an aqueous solution containing a surfactant as a stabilizer [14]. They are frequently used as a formulation strategy for cosmetic and therapeutic materials, to control the release, increase the bioavailability, and enhance the stability of active compounds [14,15]. Given the similarity of their composition to stratum corneum components, they are particularly useful nanocarriers for skin applications. SLNs can be freeze-dried before being incorporated into tablets for later reconstitution. Freeze-dried SLNs combined with surfactants can improve both the tablet formation and their subsequent dissolution [16,17]

Third, key ingredients that can promote the skin microbiota can be identified. Fatty acids are important for the maintenance of normal skin barrier function, reducing the loss of moisture from the skin, maintaining the metabolism of skin cells, and contributing to the skin microbiota [18,19]. Short-chain fatty acids (SCFAs), enzymes, peptides, and vitamins, described as postbiotics, can be collected as metabolic by-products secreted by live bacteria/yeast or released after bacterial/yeast lysis. They can positively affect microbiota homeostasis, thus affecting specific physiological and immunological reactions [20,21], and may be useful in preventing viral infections caused by the transmission of viruses to the hands after touching contaminated surfaces [20]. Prebiotics are non-digestible dietary fibers, including mainly carbohydrates that provide nutrition to selectively stimulate the activity and growth of normal skin microbiota. Some prebiotics, such as oligosaccharides and xylitol, can also promote skin health [22]. 

A fourth consideration is that the formulation development should be based on sustainability and promoting good environmental principles. There is a drive to reduce water in pharmaceutical, cosmetic, and personal care formulations through the manufacture of anhydrous or waterless products, such as sticks and tablets. These may be more economical and beneficial for skincare and the environment because they are less prone to microbial contamination, have a longer shelf life, need fewer preservatives, need less packaging, and are more potent due to a higher concentration of active ingredients than water-based cosmetics/pharmaceuticals [23]. 

Applying the considerations discussed above, the aim of this study was to develop a waterless hand sanitizer with easy topical application and storage for skin barrier maintenance, virus protection, and fast microbiome reestablishment. The postbiotics used were tyndallized *Lactobacillus* sp. and a combination of long-, medium-, and short-chain fatty acids [24], niacinamide, and alpha-glucan oligosaccharide. *Polymnia sonchifolia* root juice and maltodextrin were selected as prebiotics. A glycyrrhizin salt (dipotassium glycyrrhizinate) was included, as it has been reported to attenuate skin inflammation (such as pro-inflammatory cytokines Il-1b and Il-6) [25,26], thus contributing to skin barrier repair [27].

For a sustainable product, these bioactive compounds were combined in a handrub tablet that was easily dispersed by contact with water (0.5–1.0 mL) and did not require rinse-off after use. It was applied on hands as a leave-on product to continually sanitize the skin and feed the resident microbiota. The tablet was prepared by associating hydrophilic prebiotics with hydrophobic postbiotics that are pre-entrapped in SLNs. These SLNs can be attached to the skin surface due to the cationic properties of quaternary ammonium compounds, which also exert antiseptic activity against temporary viruses such as SARS-CoV-2, pathogenic bacteria, and fungi, together with the lipophilic postbiotics in the core of the nanoparticle responsible for the skin homeostasis recovery and overall skin health.

The approach taken in the study is shown in the flowchart (Figure 1).

## 2. Materials and Methods

### 2.1. Materials

Cetyltrimethylammonium chloride (Sunquart CT-50^®^), palmitic acid, and stearic acid were purchased from Aqia (Guarulhos, Brazil); poloxamer 188 (Kolliphor^®^ P 188) and polyvinylpyrrolidone K30 (Kollidon 30^®^) from BASF (São Paulo, Brazil); menthol from Mapric (São Paulo, Brazil); fragrance (Sunflower Fav-310939) from Fav 105 (Diadema, Brazil); silicon dioxide (Aerosil^®^) from Evonik (São Paulo, Brazil); sodium croscarmellose (Solutab^®^), magnesium stearate, and microcrystalline cellulose PH101 (Microcel^®^ PH101) from Blanver (Taboão da Serra, Brazil); xylitol, lauric acid, and caprylic acid from Química Anastácio (São Paulo, Brazil); hydroxypropyl cellulose (Klucel TM^®^) from Ashland (São Paulo, Brazil); and glyceryl monostearate from Labsynth (Diadema, Brazil). Alpha-glucan oligosaccharide, *Polymnia sonchifolia* root juice, maltodextrin, *Lactobacillus* (Ecoskin^®^), and alpha-glucan oligosaccharide (Bioecolia^®^) were from Solabia (Maringá, Brazil); niacinamide (Niacinamide PC^®^) from DSM (São Paulo, Brazil); glycyrrhizinate dipotassium from Cosmotec (Guarulhos, Brazil); and valeric acid, butyric acid, and oleic acid from Sigma-Aldrich (São Paulo, Brazil)**.** All the reagents and solvents were analytical grade and were purchased from commercial suppliers.

### 2.2. Development of Tablet Base Formulation

Design of Experiments (DoE) was used to identify which factors were relevant to the development of the base formulation to prepare tablets for the loading of the lipid nanoparticles. Firstly, we established which set of features the base tablet (formulations only prepared with excipients) required to fulfill the aim of being a solid dosage form that rapidly disintegrated after absorbing a small volume of water, leaving a smooth and easy-to-apply on hands fluid gel. The tablets should show minimal hardness, only sufficient for easy handling, and a high capability to absorb water and lose their structure quickly in contact with water (0.5 mL). Hence, tablets should show a high swelling rate and low disintegration time. Additionally, the height and weight of the tablets were monitored.

The DoE was performed using Minitab 18.1 software for planning and data analysis. A 2^3^ factorial planning added off a central point in triplicate was used, resulting in eleven formulations. In this model, 3 formulation factors (independent variables) in 2 levels were considered: 1—the type of diluents used, microcrystalline cellulose PH101 (MCC) or pregelatinized starch (PGS) (in the levels 1:0, 0:1, and 0.5:0.5 as central point); 2—the type of super-disintegrants used, sodium croscarmellose (SCC), or sodium starch glycolate (SSG) (in the levels 1:0, 0:1, and 0.5:0.5 as central point); 3—the concentration of super-disintegrants in formulation (in the levels 10, 20% *w*/*w* and 15% *w*/*w* as central point). In addition to the diluents and super-disintegrants, the formulations were compounded with colloidal silicon dioxide and magnesium stearate as lubricants (Table 1).

The tablets were obtained by compression of the formulations previously weighed and blended to produce 5.0 g powder mixtures. The powder mixtures were fractioned in portions of 100 mg and transferred to a stainless-steel matrix (8.0 mm diameter) with the bottom blocked with a stainless-steel plate attached to the matrix. With the powder formulation inside, a punch was positioned into the hole of the matrix and compressed up to 50 psi in a manual hydraulic press (Wika, Barueri, Brazil). The tablets obtained were assessed for hardness, disintegration time, swelling rate, weight, height, and diameter. All these parameters were considered as dependent variables in the DoE study. The response surface methodology was used to identify the critical parameter of the formulation and possible interactions between them.

#### 2.2.1. Characterization of Tablet Base

The formulations of the tablet base were evaluated using weight variation, hardness, diameter, thickness, swelling rate, and disintegration time as parameters. All tests were performed using a minimum of 3 tablets from each formulation. Dimensions of the tablet (diameter and thickness) were determined using a micrometer (External Micrometer 436.2 series, Starret, Itu, Brazil). The weight variation was determined by weighing each tablet individually on an analytical balance (Shimadzu, AY 220, Barueri, Brazil) to ensure weight uniformity. The hardness of three tablets was measured using a durometer (298 ATTS, Nova Ética, São Paulo, Brazil), which recorded the force (N) necessary to break the tablets under diametric compression. The acceptance criterion for breakage was a pressure of not less than 10 N [28]. Swelling rate was determined by the difference (%) between the weight of one tablet after and before contact for 30 s on the center of a wet sponge in purified water, placed in a Petri dish. Additionally, disintegration time was determined as the time required for the tablet to completely disrupt in 5 mL purified water under orbital stirring at 200 rpm (Solab, Incubator Shaker SL 222, Piracicaba, Brazil). 

The data recorded were used in the response surface methodology to analyze the effect of the factors in the response parameters, aiming to identify those that are more important for the set of features intended.

### 2.3. Preparation of Solid Lipid Nanoparticles (SLNs)

SLNs were prepared with the concentration ranges shown in Table 2, based on the patent application (Patent Number BR 10 2021 020202 5) [29] by emulsification of the aqueous phase and oily phase followed by high-pressure homogenization (Nano DeBee, Model 30-4, Bee Instruments, Easton, MA, USA). Aqueous and oily phases were weighed separately and heated to 70 °C. The aqueous phase was dispersed into the oil phase and stirred for 10 min. About 5 mL of each formulation was separated and characterized. In addition, the emulsion was treated in closed continuous flow for 10 min under pressure of 10,000 psi and counter-pressure of 1500 psi [15,30].

#### 2.3.1. Characterization of Solid Lipid Nanoparticles

Laser diffraction (Cilas model 1190) was used to characterize the particle size properties of the SLN dispersions [28,29]. This technique allows the evaluation of particles in the range of 0.04 to 2500 µm. The algorithm that considers Mie’s Light Diffusion Theory, available in the software of the equipment, was used to calculate the size parameters, using 1.46 and 1.33 as the refraction indexes of lipid nanoparticles and water, respectively. The values of the diameter related to 10% (d10), 50% (d50), and 90% (d90) of the particle population were recorded and used to calculate the SPAM of the samples (SPAM = (d90 − d10)/d50), giving information regarding the size dispersity. Additionally, the mean diameters (dm) were recorded in two ways: volume distribution and number distribution. The volume distribution considers the volume occupied by one specific population of particles compared to the volume of the totality of the particles analyzed, establishing the size distribution related to volume. It gives an indication of the yield of the process in providing particles in a size range. The number distribution gives an indication of the capability of the process to generate particles in a given size range because it measures which range of particle size is more frequent (in units of particles, number) in the nanoparticles’ dispersion. Thus, if the volume and number distributions of a sample are very different, a wide range of size particles is present, indicating an unacceptable size distribution. Consequently, the uniformity ratio (UR = [dm-volume/dm-number]) was calculated. A UR value of 1.0 represents a perfectly uniform size distribution, and, ideally, the UR should be as close to 1.0 as possible. A UR greater than 10 indicates an extremely heterogeneous sample, whereas a moderately heterogeneous size distribution is characterized by UR values between 5 and 10. For size analyses, all measurements were performed in triplicate, and values of mean, standard deviation, and relative standard deviation were calculated.

#### 2.3.2. Prepare of SLN Dispersion as a Solid Component 

To disperse the SLNs as a solid component in the tablet base formulation, a dispersion of nanoparticles (20 g) was added to the diluent selected (20 g) in the DoE study. The procedure used was similar to that used in wet granulation, with the SLN dispersion (a liquid) being poured onto a powder formulation (a solid) under blending. The process was performed in a bench-batch; hence, no pilot or industrial equipment was used. After homogenization, the samples were frozen at −20 °C for 24 h and freeze dried for 48 h, under a vacuum at 40 µmHg (5.26 × 10^5^ atm) and a condensation chamber temperature of −60 °C. The lyophilized formulation was packed in airtight glass containers and stored at 5 °C. The residual humidity was evaluated in a heating balance with a halogen lamp (MAC 210, Radwag, Random, Poland), and the final point of the analysis was automatically determined by the stabilization of the sample’s weight.

#### 2.3.3. Thermal Analysis: Differential Scanning Calorimetry (DSC) and Thermogravimetric Analysis (TGA) 

DSC and TGA measurements were performed using a simultaneous thermal analyzer DSC/TGA: the Discovery SDT 650 from TA Instruments of the Laboratório de Materiais Híbridos (LMH), Unifesp, Campus Diadema, SP. DSC/TGA curves were obtained at a heating rate of 10 °C min^−1^, in the temperature range from room temperature to 700 °C, under a dynamic synthetic air atmosphere (50 mL min^−1^), using an alumina crucible (90 µL). Samples of the acids butyric, valeric, caprylic, and oleic were assessed separately. Furthermore, a sample containing all components of the oil phase was obtained and divided into two parts. One part was kept at room temperature, and another was frozen at −20 °C and lyophilized for 24 h, under vacuum at 0.027 mbar. Similarly, the dispersion of SLN (F6) was prepared as previously described and divided into two parts, one kept at room temperature until analysis and another lyophilized as described before. The thermograms were recorded to evaluate thermic events related to physical or weight changes.

#### 2.3.4. Desorption of SLN from Lyophilized Powder Mixture

Desorption studies were performed to evaluate the ability of the SLNs to be released from the Microcrystalline cellulose PH101 into the aqueous medium, where they would be free to interact with the skin surface and to play a role as a nanocarrier of postbiotics. Thus, 500 mg of each lyophilized sample was added to a 20 mL vial containing 6 mL of purified water. The samples were kept under magnetic stirring for 1 h at room temperature, then centrifuged at 5000 rpm for 10 min to separate the particles of diluent dispersed in the aqueous medium. The supernatant was collected and analyzed by laser diffraction to assess the profile of particle size distribution and its corresponding SLN dispersion. 

### 2.4. Development of the Handrub Tablet

The handrub tablets were prepared using the best-performing tablet base (F6) and the previously prepared and characterized freeze-dried SLNs, with other additional ingredients shown in Table 3. Prebiotics consisted of niacinamide, alpha-glucan oligosaccharide, *Polymnia sonchifolia* root juice, and maltodextrin. Postbiotics used were tyndallized *Lactobacillus* sp. and the short-chain fatty acids contained in the SLN. The concentration ranges shown are as specified in the patent application [29].

The handrub tablets were manufactured by direct compression, using a rotary press simulator (Mini Express LM 8, Lemaq, Diadema, Brazil) with a set of circular punches and dies, 8 mm in diameter. All powders were previously mixed and passed through a 45-mesh sieve. The resulting tablets had a mass of about 100 mg.

A synthesis scheme by steps for development of handrub tablet is shown in Figure 2.

#### Characterization of the Handrub Tablet

The handrub tablet was evaluated by weight variation, hardness, diameter, thickness, swelling rate, and disintegration time, as previously described in Section 2.2.1.

### 2.5. Safety and Efficacy Evaluation of the Handrub Tablet

#### 2.5.1. In Vitro Cell Preparations

The cytotoxicity was evaluated in a cell monolayer, with spontaneously immortalized normal human keratinocytes cultured with DMEM (Dulbecco’s Modified Eagle’s Medium) (Thermo Fisher Scientific, São Paulo, Brazil) modified to contain 4 mM L-glutamine, 4500 mg/L glucose, 1 mM sodium pyruvate, and 1500 mg/L sodium bicarbonate and fetal bovine serum to a final concentration of 10%., in an incubator at 37 °C and 5% CO_2_ [31]. 

Human skin fibroblasts cells were maintained in the same conditions, for 48 h, to analyze the potential of the samples in collagen type 1 synthesis, and for 24 h for antioxidant activity analysis.

To prepare a human skin equivalent model, fibroblasts were embedded in a protein matrix. On the surface of this matrix, keratinocytes were added to form the dermis and epidermis for the equivalent skin model [32]. The human skin equivalent preparations were cultivated to allow stratification of the epidermis and then used for the study of virucide activity and gene expression analysis, as described by Carlson et al., 2008 [33]. 

In addition, for the virucide activity assay, VERO cells (C1008) (ATCC CRL1586, Rio de Janeiro Cell Bank, Duque de Caxias, Brazil) were used, cultivated with DMEM (Dulbecco’s Modified Eagle’s Medium) Low Glucose (Thermo Fisher Scientific, Brazil) with 2 mM Glutamine and 10% of fetal Bovine Serum in an incubator at 37 °C and 5% CO_2_ and manipulated inside a laminar flow hood [34]. At passage 06 after thawing, the cells were distributed in plates suitable for monolayer culture [31].

In all analyses, each tablet was diluted in 1 mL of growth factor-reducing culture medium with a final concentration of 100 mg/mL. A minimum of 3 replicates of each concentration were tested, and control cells were exposed to the culture medium only.

#### 2.5.2. In Vitro Cytotoxicity

Initially, the handrub tablets were evaluated using the neutral red assay, as a safety screening assay. After cell cultivation, at the fourth passage after thawing, the cells were distributed in a culture plate. The solutions containing the control group and the handrub tablet were applied to the cell culture, 100 μL per well in triplicate for each group, followed by incubation in an oven at 37 °C and 5% CO_2_ for 24 h. After washing, cell viability was analyzed with the neutral red indicator, 100 μL per well. After incubation, absorbance was determined at 540 nm in a spectrophotometer [31].

#### 2.5.3. In Vitro Virucidal Activity

To check the protective action against alpha-coronavirus (CCoV-VR809), a member of the SARS-CoV-2 family, in vitro tests for antiviral activity were performed on a VERO cell monolayer. Handrub tablets (100 mg) were diluted in 6 mL (16.6 mg/mL), considering the same concentration for use in the hands area (600 cm^2^). The samples were applied over a concentration of 10^5.0^ particles of alpha-coronavirus (CCoV-VR809). The sample was in contact for 1 min and 120 min on monolayer cells. This inoculated cell culture was evaluated according to changes in morphology, which are characterized by the cytopathogenic effect caused by the tested viruses. A comparison of the negative control group (no virus) with the viral control group was performed, and the application of sample was performed to assess the presence of viral replication and comparison with viral titration performed, which shows the reduction in the number of infectious viral particles in logarithms. 

For the human skin equivalent assay, after epidermis stratification, the protocol for infecting the skin with viral particles was followed. A solution with a previously established number of viral particles in a concentration of 10^4.0^ particles of alpha-coronavirus (CCoV-VR809) was applied on human skin equivalent. Five minutes after the application of 70% alcohol group and the handrub tablet, samples were collected in culture medium, and later the quantification of viral particles was performed. A negative control group was performed, in which there was no application of viral particles on the equivalent skin.

The results are expressed as a reduction in logarithms of the number of viral particles and correlated with the observed percentage.

#### 2.5.4. In Vitro Gene Expression

For gene expression analysis, the samples, at a concentration of 0.01 mg/mL, were applied to the keratinocyte cell culture, as a screening of the potential benefits of the handrub tablet. The keratinocyte messenger RNA was then extracted with trizol, and the quantity and purity assessed. It was converted into a complementary DNA strand, and real time Polymerase Chain Reaction (qRTPCR—StepOne Plus from Applied Biosystems, Waltham, MA, USA) analysis was used to quantify the level of expression of moisturizing markers (aquaporin-3 (AQP-3)), skin barrier markers (involucrin and filaggrin), and the cytokines interleukin-6 (IL-6) and interleukin-8 (IL-8)). In addition, the expression of sirtuin-1 (SIRT-1) and forkhead box O3 (FoxO3), two important senescence markers, was evaluated. The GAPDH marker was used as an endogenous control.

In the human skin equivalent evaluation, the solutions containing the control and sample groups were applied to the skin equivalent model surface, at a concentration of 10.0 mg/mL, every other day for seven days. The extraction of messenger RNA with trizol was then performed, evaluating its quantity and purity of each group. The PCR technique was again used to evaluate the makers noted above, plus interleukin-1 (IL-1), interleukin-17 (IL-17), and collagen type 1 synthesis. In this assay, sodium dodecyl sulfate (SDS) was used as negative control for IL-1, IL-6, and Il-8; phorbol-12-myristate-13-acetate (PMA) for IL-17; D-Pantenol for AQP-3, involucrin, and filaggrin; and resveratrol for SIRT-1, FoxO3, and collagen type-1 synthesis, as a positive control. The medium culture was considered as basal control in all analyses.

#### 2.5.5. In Vitro Antioxidant Activity

To evaluate the antioxidant potential, human fibroblast cells were used, maintained in culture with DMEM (Dulbecco’s Modified Eagle’s Medium) with the addition of supplements at 37 °C and 5% CO_2_. Each handrub tablet was diluted in 1 mL of growth factor-reducing culture medium with a final concentration of 100 mg/mL. Cell culture was used as a control group and Trolox as a positive control. The solutions containing the control, positive control, and sample groups were applied to the cell culture, 100 μL per well in quintuplicate for each group, followed by incubation in an oven at 37 °C and 5% CO_2_ for 24 h. After washing, the presence of free radicals was analyzed using the CM-H2DCFDA probe with an excitation wavelength of 495 nm and emission of 525 nm. 

#### 2.5.6. In Vitro Collagen Synthesis

Control and handrub tablet samples were also used to evaluate the collagen type 1 synthesis in fibroblast skin cells and in human equivalent skin with resveratrol as a positive control. The amount of collagen produced by fibroblasts and human equivalent skin was evaluated using a picrosirius red marker. Samples were read with absorbance at 540 nm. 

#### 2.5.7. In Vivo Efficacy: Microbiome Analysis

Fifteen healthy female participants, between 18 and 60 years old, were selected for hands metagenomic analysis, with ethical approval granted by Centro Universitário Padre Anchieta Human Research Ethics Committee (Approval No. 4.766.469).

They were separated into 3 groups: 5 volunteers with 70% alcohol plus handrub tablet (G1), 5 volunteers with handrub tablet (G2), and 5 volunteers with 70% alcohol (G3). 

For the 70% alcohol application, the volunteers were instructed to apply enough for complete coverage of the hands, including the palms, the backs of the hands, and between the fingers. The volunteers assigned to use the handrub tablet were instructed to place the tablet in the palm of their hands and apply 20 drops of water to disintegrate it, then spread it over their entire hands, as above. They were instructed to not rinse the product off. The group with 70% alcohol plus handrub tablet were asked to first apply the 70% alcohol to their entire hands, allow it to dry, and then apply the handrub tablet as above. All volunteers used their designated products for seven days, three times a day.

The participants were informed that they must not use any antiseptic before attending the clinical trial. On the first day, upon arrival at the research center, after signing the informed consent form, a sample of the microbiome of the hands was collected (T1), without using any product. Afterwards, the participants were invited to wash their hands with neutral soap and apply the product from their test group (G1, G2, and G3). Five minutes later, the second sample (T2) was performed and again after two hours (T3). For seven days, the volunteers used products assigned to them. On the eighth day, a new sample was collected upon arrival at the research center (T4). The participants washed their hands with neutral soap and after two hours the last collection was carried out (T5).

From the collected swabs [35], metagenomic analysis (MiSeq Illumina Platform) was performed to evaluate the types as well as the variability of the bacteria. The presence of greater amounts of saprophytic bacteria, such as actinobacteria and proteobacteria, and greater variability of species are indicative of microbiome balance.

Samples were analyzed by sequencing the 16S rRNA gene V3–V4 region using MiSeq (Illumina™, San Diego, CA, USA), and, from bioinformatics analysis, qualitative and quantitative analyses of the observed results were performed. To assess the balance of the microbiome, the abundance of microorganisms identified in the collections was evaluated, and a comparison between groups and collection times was performed. 

### 2.6. Data Analysis and Statistics

The Design of Experiment 2^3^ factorial was performed by Minitab^®^ 18 Statistical Software, with factors considered significant when *p* < 0.05.

The in vitro data were evaluated using Microsoft Excel software (Microsoft Office 365). For the analysis of the data obtained by RTq-PCR, the analysis from the 2^−∆∆Ct^ was used for graphical representation of the relative expression by fold-change and statistical analysis from the ∆Ct data. Statistical analysis for comparison between groups was performed using the Student *t* test and a statistical significance level considered lower than 0.05.

## 3. Results

### 3.1. Characterization of Tablet Base

With the aim of developing a fast-disintegrating tablet that generated a dispersion that was smooth and easy to apply, 11 different formulations of tablet base were prepared as described above, using a manual hydraulic press. The experimental characteristics of the base formulations are shown in Table 4. 

The results obtained were submitted to Response Surface Regression for the model as a whole and linear, square, and 2-way interactions for the independent variables. The test was run for each dependent variable shown in Table 1. The results of *p*-values and lack-of fit are shown in a table in the Appendix A. The four parameters of weight, height, disintegration time, and swelling showed *p*-values lower than 0.05 for the model, allowing identification of the factors and their interactions that play an important role in the tablet properties. It is important to note that since it was not possible to measure the hardness with high precision, tablet weight was considered a surrogate parameter for tablet resistance. As the amount of powder weighed was always 100 mg, variations in this weight might indicate that the tablets are not well-formed, causing material loss during compression or in handling. Therefore, weights closest to 100 mg were taken to indicate tablets with better resistance characteristics.

The Pareto charts of the standardized effects (Appendix A) illustrate how the independent variables influenced a given feature of the tablet base formulations. It was observed that Diluent Proportion is the main factor to influence the parameters’ mean weight, height, and disintegration time, whereas the factor Disintegrant Proportion assumes major importance in determining the swelling parameter. Curiously, the concentration of disintegrant does not play a major role in the properties of the tablet base formulations, possibility because the lower concentration used is sufficient to promote the highest effect of disintegration. The contour plots (Appendix A) showing the interactions of factors Disintegrant Concentration vs. Diluent Proportion (color of the bands are practically vertical) and Disintegrant Proportion vs. Disintegrant Concentration (color of the bands are practically horizontal) indicate that no important variation was observed for disintegration time in the range of 10 to 20% *w*/*w* disintegrant concentration.

The contour plots (Appendix A) show how a tablet feature can vary considering the interaction of two factors, and this information provides information to optimize the formulation to reach the main aims of lowest disintegration time and lowest swelling. While the results for disintegration time was expected, the swelling feature resulted in combined analysis with disintegration time, showing that the disintegration time also rose when swelling achieved the highest values, which was considered an undesirable outcome. This might be attributed to the gelation of the formulation keeping the formulation particles together. Furthermore, the greatest weight (as a resistance feature) and greatest height (as a parameter able to interfere in the density of tablets) were set as parameters for optimization.

The set of parameters and results of optimization are shown in the Appendix A. The optimization reaches a composite desirability index of 0.9808, which was considered adequate since the highest possible value is 1.0. The composition of the optimized formulation was equivalent to F6, previously tested during the DoE experiment, which was adopted as the optimal tablet base formulation to receive nanoparticles and other pre- and post-biotics.

### 3.2. Characterization of SLNs

In extensive preliminary investigations, we performed nanoparticle dispersion with and without high pressure homogenization and found that the particle size distribution was much more uniform with high pressure homogenization. 

The values of d10, d50, d90, and dm obtained from laser diffraction size analysis of SLNs and the calculated values of dispersity (Spam = (d90 − d10)/d50) and uniformity ratio (RU = (dm passing volume)/dm passing number) are shown in Table 5. The particle size distribution with high pressure homogenization, expressed in passing volume, is shown in Figure 3. As a main result, we showed mean size (310 nm) and Spam (2.43). This study prompted us to use SLNs for free fatty acid (postbiotics) encapsulation, to protect the fatty acids from degradation, and to minimize the characteristic odor of short-chain fatty acids used as postbiotics. More information is given in the next section.

#### Thermal Analysis of SLNs

As the SLNs were required to remain solid on application to the skin, it was essential to establish whether this was the case by testing at body temperature. Furthermore, some of the active compounds added in the formulation were liquid fatty acids, which evaporate substantially at room temperature, even though their boiling point is higher than 160 °C. 

For example, in the analyses of the liquid fatty acids, we observed that butyric acid lost weight from around room temperature, whereas valeric and caprylic acid started to lose weight at around 70–80 °C, the temperature of the nanoparticle manufacturing process (Table 6). Oleic acid, a long-chain fatty acid, starts to lose weight at 134 °C, although the first event was observed at 101 °C, not related to weight loss. As oleic acid is an unsaturated fatty acid, its chemical structure can assume two forms, Cis and Trans, which can interconvert on warming [36]. Two other events were observed for oleic acid, one related to evaporation (Tonset = 287 °C) with weight loss recorded to almost 90% (DTG peak at 325 °C) and another event of weight loss related to degradation of the residue. The event related to evaporation had a peak at 331 °C in the DSC curve, which is different to the previously reported boiling point (360 °C) for oleic acid [37]. The same might be observed for the other three acids, even though the difference related to evaporation temperature determined in this work and boiling point in the literature varied between 2 and 7 °C. 

Differently from the in natura liquid fatty acids, the mixture of those with solid fatty acids and other raw materials (oil phase non-lyophilized) was able to increase the temperature where the weight loss took place above 97 °C. This feature was observed for the other samples, where a mixture of liquid and solid compounds was used (oil phase lyophilized and lipid nanoparticle lyophilized). This result is interesting since the encapsulation of liquid short-chain fatty acids in lipid nanoparticles can be an alternative to reduce the unpleasant odor of these compounds in formulations for microbiota rebalance. Furthermore, for three of these mixtures the events related to weight loss effectively happened in two steps, the first with T_onset_ varying between 180 and 250 °C, and the second with T_onset_ between 380 and 410 °C. 

An event was present on the thermograms of the samples oil phase non-lyophilized and lyophilized at 46 and 49 °C (Figure 4), respectively. This event might be linked with an endothermic peak related to the melting of the lipid mixture resulting from the oil phase. Despite the same oil phase composition being present in the lipid nanoparticle samples, when analyzed as the aqueous dispersion, it was not possible to observe this peak since water evaporation is a dominant event at that temperature range. In the lyophilized lipid nanoparticle sample, the water was eliminated by freeze-drying; hence, the event of melting should be seen. However, no event was observed in the range of 45 to 55 °C indicating the absence of any crystalline structure in the lipid nanoparticles after lyophilization. This finding suggests that the structure of the lipid nanoparticles is mainly amorphous, due to the liquid fatty acids used in the formulation, preventing the crystallization of the solid fatty acids and consequent expulsion of the liquid acids from the nanoparticle lipid matrix.

### 3.3. Release of Freeze-Dried SLNs

The freeze-dried SLNs had a residual humidity lower than 3% and an average particle size lower than 500 nm, thus being considered suitable for incorporation into the tablet. The size and size distribution values are shown in Table 7. Particle size distribution with high pressure homogenization obtained by laser diffraction technique, expressed in passing volume, is shown in Figure 4.

### 3.4. Characterization of the Handrub Tablet

DoE analysis showed that F6 was the best-performing base tablet formulation, in terms of the parameters average weight, tablet thickness, disintegration time, and swelling rate, which were significantly influenced by the independent variables shown in Table 1. F6 was therefore chosen for the development of the handrub tablet. The final product, obtained by incorporating the bioactive ingredients, including pre-/postbiotics and SLNs into the base tablet F6, was then taken forward for safety and efficacy evaluation. The physical characteristics of the final handrub tablet are shown in Table 8.

### 3.5. In Vitro Cytotoxicity

Cell viability was assessed by treating keratinocyte monolayer preparations with varying concentrations of the handrub tablet diluted in DMEM for 24 h. Results, expressed as a percentage normalized to the control group at 100%, are shown in Figure 5. Increasing concentrations from 0.1 mg/mL were considered cytotoxic [38]. Treatment with a concentration of 0.01 mg/mL resulted in cell viability of greater than 80%. This concentration was regarded as non-cytotoxic and was therefore selected for the later gene expression studies. 

### 3.6. In Vitro Virucidal Activity

The antiviral activity of the handrub tablet was evaluated by exposure of CCoV-VR809 viral particles inoculated onto VERO cell cultures (1 and 120 min) and in human equivalent skin (5 min). To the viral control group, the presence of 10^5.0^ and 10^4.0^ of the particles was quantified, respectively, in the monolayer cells and human equivalent skin, to demonstrate successful contamination with CCoV-VR809. The handrub tablet decreased the viral particles by 99.99% in VERO cell culture analysis. In human equivalent skin, the handrub tablet caused a reduction of 99.96% of viral particles, a better performance than 70% alcohol (99.9% reduction). In the negative control group, no viral particles were detected. These results are summarized in Table 9.

### 3.7. In Vitro Gene Expression

The gene expression of the skin moisturizing and barrier function markers aquaporin-3, filaggrin, and involucrin in response to handrub tablet treatment were determined by RTq-PCR. Figure 6A–C shows small but non-significant increases in expression of these markers after handrub treatment, demonstrating that the handrub tablet was able to maintain the skin barrier without damaging it.

Although useful in the evaluation of safety and efficacy of cosmetics and pharmaceutical products, fast in vitro assays in monolayer cells are unable to reproduce the full range of normal skin responses. A human skin equivalent model, with a skin barrier and better-defined epidermal and dermal layers, can more closely replicate the interactions and responses occurring in living skin [32].

In our human skin equivalent model, there was no significant increase in involucrin expression compared to control. However, significant increases in aquaporin-3 (1.41-fold-change, or 40.8%) and filaggrin (2.78-fold-change, or 178.2%) expression were observed after handrub tablet exposure, compared to control (Figure 7). The handrub tablet showed similar results to the positive control (D-panthenol) in the involucrin assay. This result may be related to the expression of involucrin occurring in the stratum granulosum in native human skin but in the stratum spinosum in the human equivalent skin model, with the culture conditions that need to be adjusted [35]. These findings in a human skin equivalent model with barrier function, unlike the monolayer model, demonstrate that use of the handrub tablet can lead to improved skin hydration and barrier function through aquaporin and filaggrin increases, respectively.

To evaluate the effect of handrub tablet on the inflammatory process, gene expression of the interleukins IL-6 and IL-8 were quantified in monolayer cells. The results are shown in Figure 8. IL-6 expression is increased in epidermal layers after barrier disruption and reduces the amount of ceramide in the stratum corneum [27]. A slight but non-significant increase for both inflammatory markers was seen in response to the handrub tablet compared to control, demonstrating that the tablet did not initiate an inflammatory process and suggesting it had no negative effect on skin integrity.

However, when the handrub tablet was applied to the human skin equivalent model, significantly reduced responses were seen with IL-1 (0.255-fold-change, or 74.51% reduction; *p* < 0.01), IL-6 (0.154-fold-change, or 82.59% reduction; *p* < 0.05), IL-8 (0.316-fold-change, or 68.44% reduction; *p* < 0.01), and IL-17 (0.111-fold-change, or 88.85% reduction; *p* < 0.01; Figure 9). In contrast, exposure to negative controls (SDS and PMA) generated an increase in these markers, with *p* < 0.01 when compared to the handrub tablet.

The expression of skin senescence markers FoxO3 and SIRT-1 was evaluated in monolayer cells (Figure 10). FoxO3 expression increased by a 1.853-fold change, or an 85.33% increase compared to baseline control (*p* < 0.05), whereas SIRT-1 expression increased by 1.498-fold, or 49.80% (*p* < 0.01).

However, in the human skin equivalent model, small, non-significant increases in FoxO3 and SIRT-1 expression were observed compared to the control (Figure 11). For FoxO3, the increase was 1.399-fold, or 39.93%, whereas SIRT-1 expression increased by 1.186-fold, or 18.57%, and the data were not significantly different in t-student analysis. No significant increases in FoxO3 and SIRT-1 expression were seen with the positive control (resveratrol) in this human skin equivalent model. 

### 3.8. In Vitro Antioxidant Activity

Antioxidant activity was assessed by the effect on free radical production in cultured human fibroblast cells. Treatment with the handrub tablet (100 mg/mL in growth factor reduced culture medium) reduced the production of free radicals significantly by 44.68% (*p* < 0.001) compared to control, as shown in Figure 12. This finding suggests that regular application of the handrub tablet may be able to counteract one of the major causes of skin ageing. 

### 3.9. In Vitro Collagen Synthesis

The effect of handrub tablet treatment on type 1 Collagen synthesis was evaluated in human fibroblast cell monolayers. A significant increase in collagen synthesis of 52.47% compared to the control group (*p* < 0.001) was found, as shown in Figure 13A.

In addition, there was a 106.98% increase in Type 1 Collagen synthesis due to handrub tablet treatment compared to control (*p* < 0.001) in the human skin equivalent model, as shown in Figure 13B. This was 11.5% lower but not significantly different to the positive control resveratrol.

### 3.10. In Vivo Efficacy: Microbiome Analysis

Having established the safety and efficacy of the handrub tablet in the in vitro tests, we proceeded to an in vivo evaluation of the handrub tablet on the skin microbiome and the extent of skin microorganism recolonization and homeostasis in human volunteers. From a swab collected from the skin of the participants, Next Generation Sequencing was performed to identify the bacteria present. A comparison was made between the three groups of participants according to the product used and the collection times to assess the impact of the products on the cutaneous microbiome of the volunteers’ hands. The main metric used was the abundance of microorganisms, which is related to the number of sequences identified per sample.

At the initial sample collection at T1, the presence of microorganisms was observed on the hands of all volunteers. This sample point was used as a basis for comparison between subsequent samples. At sample point T2, carried out 5 min after using the products, no microorganisms were identified, demonstrating that the handrub tablet and 70% alcohol had an effective antiseptic action. At T3, 2 h after application of the products, microorganisms were identified in groups G1 (handrub tablet and 70% alcohol) and G2 (handrub tablet). Participants in group G2 had the greatest abundance of microorganisms at T3. In group G3 (70% alcohol only), the presence of microorganisms was not identified in any study participant. The alcohol quickly exerts an antiseptic action when applied to the skin, and the new growth of bacteria on the skin occurs slowly after use, probably due to the sublethal effect that alcohols have on some skin bacteria [39]. This result (illustrated in Figure 14) demonstrates that the use of the handrub tablet allows the abundance of microorganisms to increase by 2 h after treatment, suggesting that its use favors skin microorganism recolonization and homeostasis.

Volunteers were instructed to use the products three times a day for seven days, after which a new microbiome evaluation was performed (at day 8, T4). A significantly greater abundance of microorganisms was observed in the comparison between times T1 and T4 in groups G1 and G2, which demonstrates that the continuous use of the handrub tablet has an impact on the balance and diversity of the microbiome with a statistical difference. These results are shown in Figure 15A,B.

After the T4 collection, participants were instructed to wash their hands, and a new collection was performed (T5) after 2 h. In this collection, microorganisms were identified in the participants of groups G1 and G2, but none were identified in the participants of the G3 group. These results again demonstrate that the continued use of the handrub tablets favors microorganism recolonization. Results are shown in Figure 16A,B.

Note that, in our study, T3 and T5 measure the resident population as the prewash with a non-antimicrobial handwash was used to remove transient microorganisms present at T1 and T4. Collectively, the abundance analysis showed that alcohol-based handrub has a destructive effect on the human skin microbiome that cannot be restored in 2 h after application, whereas the holobiont handrub tablet restores the skin microbiome within 2 h after disinfection. After repeated use, the handrub tablet can even enrich the skin microbiome and attenuate the suppressive effect on human skin microbiome induced by the 70% alcohol-based handrub. 

Previous studies have shown that over 90% of bacteria of the human skin microbiome are classified into four Phyla: Actinobacteria (52%), Firmicutes (24%), Proteobacteria (16%), and Bacteroidetes (6%) [40]. We also found the predominance of the four *Phyla*, but with different prevalence: Firmicutes (52.84%), Proteobacteria (25.82%), Actinobacteria (11.50%), and Bacteroidetes (3.12%). Our results for human hands, which represent a different scenario than other skin sites, are in line with the findings described by Edmonds-Wilson et al. [7]. It is worth noting that the predominant phylum on the hand is Firmicutes, which includes the Staphylococci family. This proportion is maintained between all the three groups and during the test period (Figure 17).

*Staphylococcus aureus*, a pathogenic bacterium, was detected in four individuals at different evaluation times. The percentage of this Gram-positive bacteria decreased with the use of the handrub tablets in all of the individuals (2.19% to 1.80%; 5.22% to 3.19%; 3.03% to not detected and 1.53% to not detected). In contrast, the percentage of the other commensal Staphylococcus bacteria was maintained or increased, like *S. epidermidis* (5% to 5.26%) and *S. warneri* (0.08 to 1.02%).

## 4. Discussion

The handrub tablets developed in this work, containing solid lipid nanoparticles, prebiotics, and postbiotics, were designed to be spread rapidly on the palms with a small amount of water (less than 1 mL) to generate a low-viscosity aqueous dispersion that stayed on the skin as a thin film, helping to fix the bioactives, and allowing them to perform their required functions. This is an eco-friendly formulation that, compared to alcoholic handrubs, is lightweight and contains little water. Thus, they have a reduced carbon footprint in terms of transportation of water from the fabrication sites to consumers, and additionally, water is saved, as they do not require rinse-off after application. 

Studies have shown that alcohol-based handrub products and surfactant-based cleansers have a broad spectrum of action against microorganisms and viruses [41,42]. Nevertheless, ABHRs have some drawbacks worth considering: (1) the frequent use of ABHRs is associated with dry skin; therefore, they are often recommended to be used with moisturizers. (2) ABHRs impair the skin barrier on people whose hands are exposed to an occlusive environment or under water immersion, such as healthcare workers who wear gloves for extended periods [43]. (3) They may induce a burning sensation on previously irritated skin [44]. (4) Alcohol may cause erythema in alcohol-intolerant people [44]. ABHRs, particularly in confined spaces, may lead to DER (disulfiram–ethanol reaction) after disulfiram treatment [45]. (5) ABHRs with high alcohol concentration are often mixed with fragrances that can be appealing to children, leading to their ingestion and associated health issues [45,46]. (6) Importantly, as shown in this study, alcoholic handrubs can disrupt the skin microbiome, which may promote skin diseases. The new skin sanitizing handrub product that was developed in this study provides superior skin protection without the aforementioned drawbacks.

The main virucidal compound in the handrub tablet is cetyltrimethylammonium chloride, a quaternary ammonium compound (QAC) [47]. Although QACs are relatively well-tolerated and have low allergic potential, they are not recommended as alternatives to ABHR due to their low activity against viral organisms compared to alcohols [2]. We therefore chose to combine the benefits of SLNs, QAC, prebiotics, postbiotics [19,22,48,49,50], and potassium glycyrrhizinate [26] to maximize the antiviral activity. Our results showed that the handrub tablet had excellent antiviral efficacy (≥99.9%) in two different in vitro model systems, offering a viable alternative to the use of alcohols.

We performed a number of gene expression studies to determine whether the handrub product caused any adverse reactions in skin and also to investigate the positive effects it may have on skin, such as improved hydration, barrier function, and anti-ageing actions. Firstly, we examined the effects on genes encoding interleukins, which are directly related to skin inflammation [51]. These cutaneous cytokines are expressed in keratinocytes and can be used as indicators of irritation or sensitization caused by exposure of the skin to external stimuli, such as the application of topical products. After handrub treatment, gene expressions of all four tested markers (IL-1, IL-6, IL-8, and IL-17) were unchanged in the keratinocyte monolayer and significantly reduced compared to the control in the human skin equivalent model, the most relevant model to human in vivo skin. From these findings, we concluded that application of the handrub tablet was unlikely to cause irritation or sensitization in human subjects.

The integrity of the skin barrier is key to healthy skin [52], and filaggrin and involucrin are important markers of this integrity. Filaggrin is fundamental for the process of terminal epidermal differentiation and has a major influence on the structure and function of the stratum corneum [49]. It is also the precursor of natural moisturizing factors (NMFs) that play an important role in the maintenance of stratum corneum hydration and skin barrier homeostasis [53,54]. Involucrin, expressed in the supra-basal region of the epidermis, is involved in the initial step in the formation of the cornified envelope that forms the corneocytes, responsible for the skin barrier function [55]. We also examined the effect of handrub treatment on the expression of aquaporin-3 (AQP3). Aquaporins, and more specifically AQP3, have been identified in mammals and shown to correlate with the ability to transport water between membranes. They are considered to be markers of skin hydration [56].

In the human skin equivalent model studies, handrub treatment caused significantly increased gene expression of filaggrin and AQP-3. This model possesses a barrier roughly analogous to that found in in vivo human skin, and our findings therefore suggest that handrub treatment can lead to improved hydration and barrier function in normal human skin. On the other hand, in the monolayer cell study, no significant differences in the expression of filaggrin, involucrin, or aquaporin-3 were seen after handrub treatment. This suggests that that other factors or pathways leading to increased expression of these markers were absent in this model that lacks a true skin barrier.

The FoxO3 transcription factor gene is expressed in many parts of the body, including skin, and is commonly known as a ‘longevity’ gene [56]. The FoxO3 protein performs a homeostatic function in skin, acting to counteract oxidative stress and managing cellular responses to apoptosis and senescence [57,58]. SIRT-1 is also known as a longevity or anti-ageing protein with the gene expressed in skin. Studies have shown activation of the SIRT-1 pathway in a MatTek cultured full thickness human skin model by resveratrol (the positive control used in our studies) [59], whereas anti-ageing treatments in normal human dermal fibroblasts also resulted in increased SIRT-1 expression [60,61]. The significant increase in FOXO3 gene expression seen here in the primary keratinocyte monolayer studies is supported by a previous report showing activation of the AMPK-FOXO3 pathway, also in primary human keratinocytes in culture [62]. FOXO3 is known to play a key role in endothelial cell regeneration and maintenance of the skin barrier [58]. Our results, showing significant increases in SIRT-1 expression in keratinocyte monolayers, are supported by previous literature reports of the involvement of SIRT-1 in epidermal differentiation [63] and anti-ageing pathways [59,60,61]. Interestingly, whereas increases in gene expression for both FOXO3 (1.4-fold) and SIRT-1 (1.2-fold) were also observed after handrub treatment in the human skin equivalent model, these did not reach statistical significance. It is still not perfectly understood how resveratrol acts in the activation of SIRT1 and FOXO3, but it is known that it does not activate this protein in native substrates, just by an indirect pathway [64]. Similarly, the positive control resveratrol did not induce significant increases (1.4-fold for FOXO3 and 1.4 for SIRT-1) in expression of these two genes.

We also observed a reduced presence of reactive oxygen species, suggesting an antioxidant skin effect, and the increase the collagen type-1 synthesis, supporting the notion that treatment with the handrub preparation can have a positive effect on the maintenance and improvement of skin health.

Of all organs in the human body, the skin is uniquely placed to interact directly with the environment in which we live. It is populated by millions of bacteria, fungi, and viruses. The recent advances in sequencing techniques have made it possible to show that the skin microbiota directly influences the health of the skin, both in defense, by producing antimicrobial and anti-inflammatory agents and preventing invasion by pathogenic microorganisms, and by acting to regulate the immune system and barrier properties. On the other hand, the dysbiosis of the skin microbiome is highly associated with the development of skin diseases [65,66,67,68,69]. 

Changes in this microbiota can lead to changes in the normal functions of the skin, which prompted us to assess whether the handrub tablet is able to prevent or create conditions to reverse such changes. In this context, a holobiont handrub containing prebiotics and postbiotics may be useful in preventing microorganism infections [49,70]. In addition, the holobiont handrub tablet contains no preservatives, many of which are likely to inhibit both pathogenic and resident microorganisms [55]. 

Interactions between the microbiome and immune systems have attracted attention. The skin microbiome has been shown to induce keratinocytes to produce IL-1, which subsequently modulates immune cells [71]. In a separate study, *P. acnes* strains from acne-affected skin were found to induce higher IL-17 levels than *P. acnes* strains from healthy skin [72]. Additionally, the *S. aureus-*α toxin can induce IL-1β production in monocytes, which may further stimulate T cells to synthesize IL-17, the proinflammatory cytokine involved in many skin diseases [73]. Interestingly, we found that the holobiont handrub tablet, which downregulated the proinflammatory cytokines IL-1α, IL-17, IL-6, and IL-8, was capable of minimizing the presence of *S. aureus* after use, thus maintaining the presence of *S. epidermidis* and other commensal Gram-positive bacteria.

## 5. Conclusions

Combining our knowledge and experience in nanotechnology-based formulation and the skin barrier, we developed an eco-friendly holobiont handrub tablet that possesses many innovative and novel attributes. Unlike alcohol-based sanitizers, our product maintained the integrity of the skin barrier, did not generate an inflammatory response, and improved markers related to skin barrier, hydration, and senescence. Regarding the skin microbiome, we conclude that the use of the handrub tablet reduces or eliminates the presence of *S. aureus*, maintaining the bacteria of the Firmicutes Phylum, such as *S. epidermidis* and *S. warneri* and other skin commensals. Unlike treatment with 70% alcohol, handrub treatment allowed recuperation of the resident skin microbiome after two hours. However, these extraordinary attributes do not come at the cost of compromised virucidal efficacy, as the handrub tablet outperformed 70% alcohol against the alpha-coronavirus. Taken together, our findings demonstrated that the handrub tablet was an effective hand sanitizer with a capacity to improve skin health and condition and was able to prevent a variety of diseases without negatively affecting the skin barrier function.

## Figures and Tables

**Figure 1 pharmaceutics-15-02793-f001:**
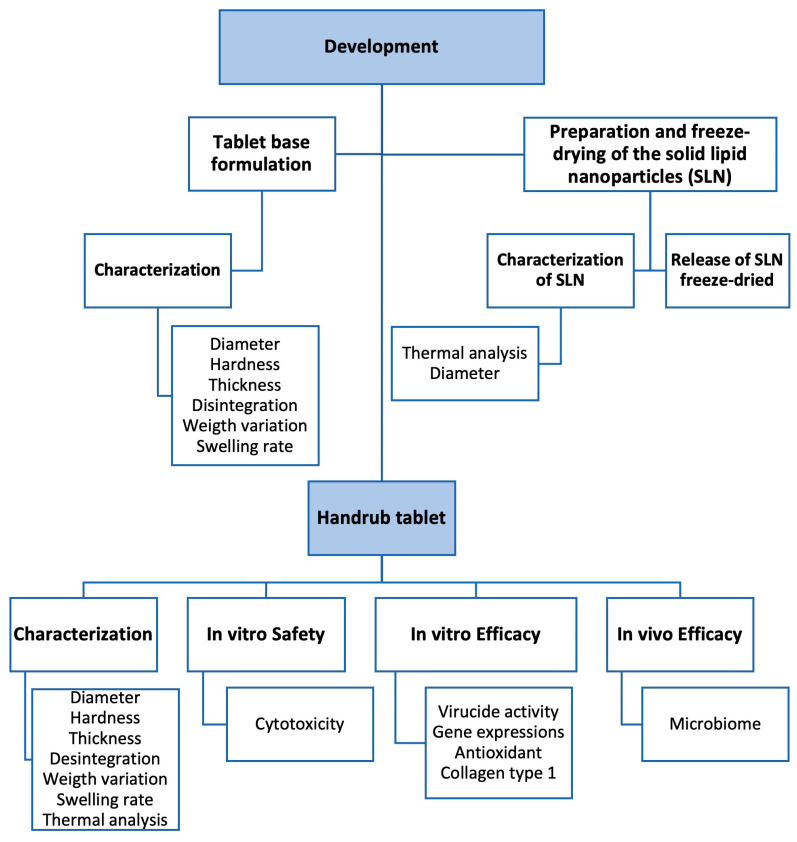
Flowchart of the development and evaluation steps of the handrub tablet.

**Figure 2 pharmaceutics-15-02793-f002:**
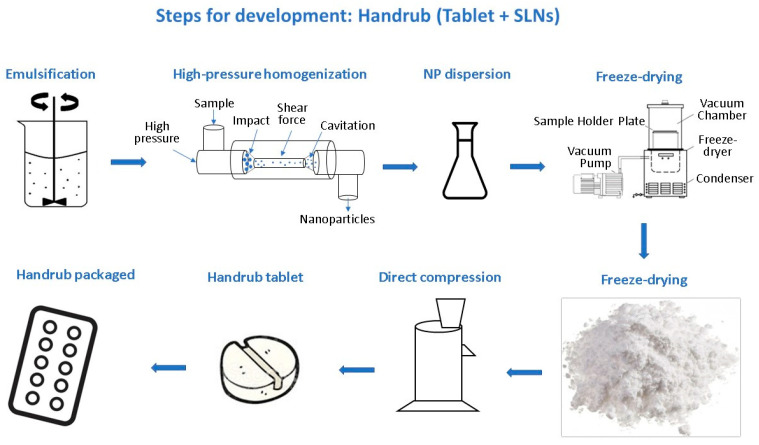
Steps for development of handrub tablet (base tablet + solid lipid nanoparticles).

**Figure 3 pharmaceutics-15-02793-f003:**
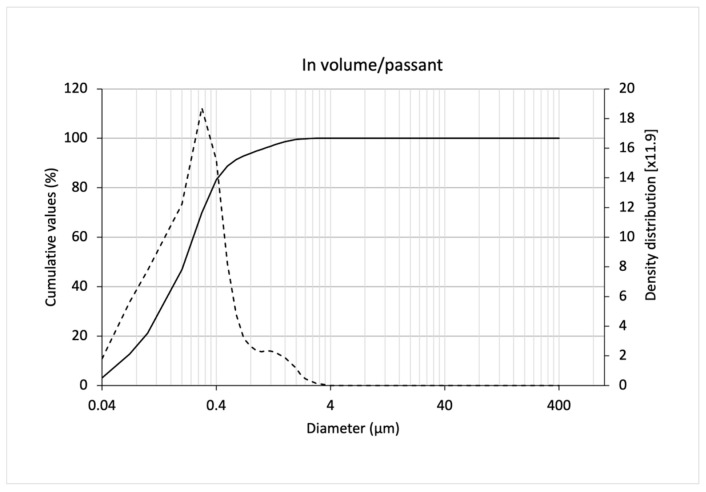
Particle size distribution of solid lipid nanoparticles obtained by laser diffraction, expressed as passing volume. (^___^) = cumulative value percentage; (---) = density distribution.

**Figure 4 pharmaceutics-15-02793-f004:**
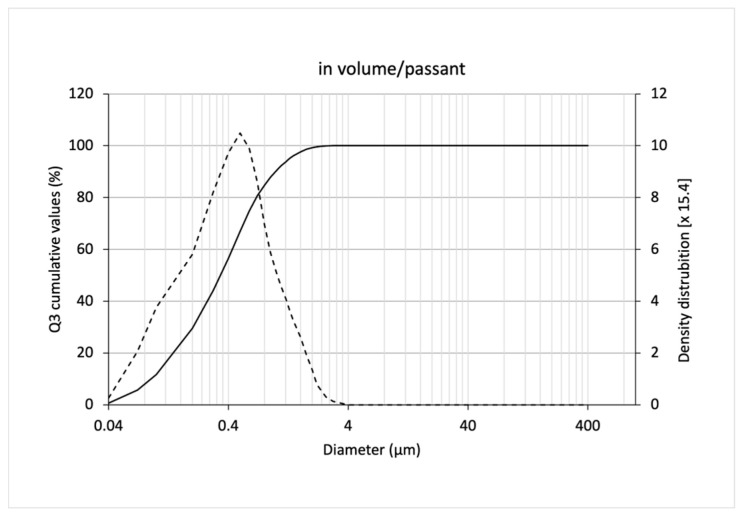
Particle size distribution of solid lipid nanoparticles and powdered diluents obtained by laser diffraction, expressed as passing volume. (^___^) = cumulative value percentage; (---) = density distribution.

**Figure 5 pharmaceutics-15-02793-f005:**
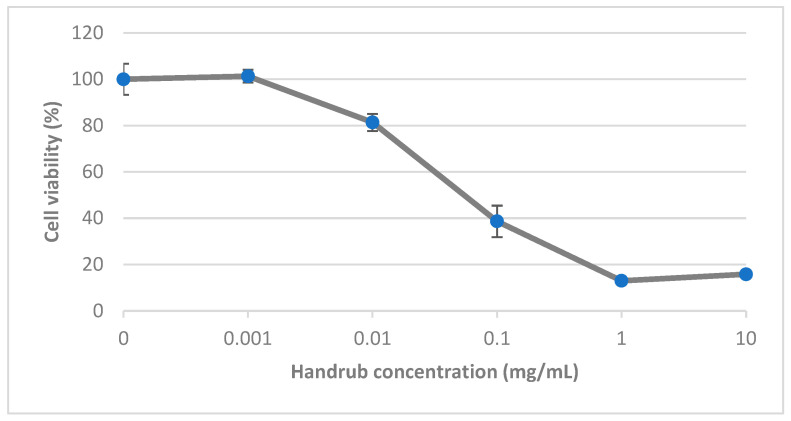
Cytotoxicity of the handrub tablet on human keratinocytes (HaCaT).

**Figure 6 pharmaceutics-15-02793-f006:**
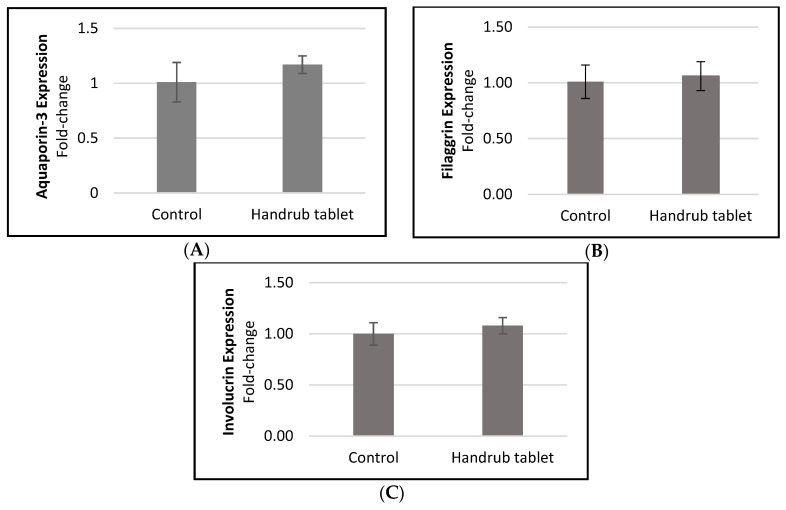
Results of the relative expression of (**A**) aquaporin-3, (**B**) filaggrin, and (**C**) involucrin, following treatment by the control and the handrub tablet in monolayer cells.

**Figure 7 pharmaceutics-15-02793-f007:**
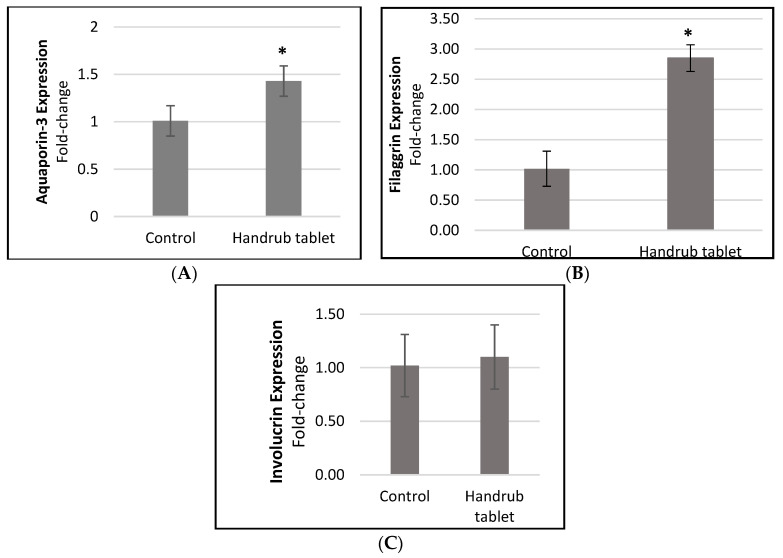
Relative expression of (**A**) aquaporin-3, (**B**) filaggrin, and (**C**) involucrin, for the control and handrub tablet in equivalent skin. * *t*-student analysis with *p* < 0.01.

**Figure 8 pharmaceutics-15-02793-f008:**
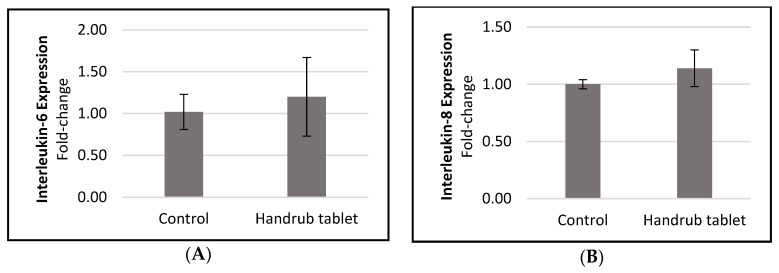
Relative gene expression of (**A**) Interleukin-6 and (**B**) Interleukin-8, following treatment by the control and handrub tablet in monolayer cells.

**Figure 9 pharmaceutics-15-02793-f009:**
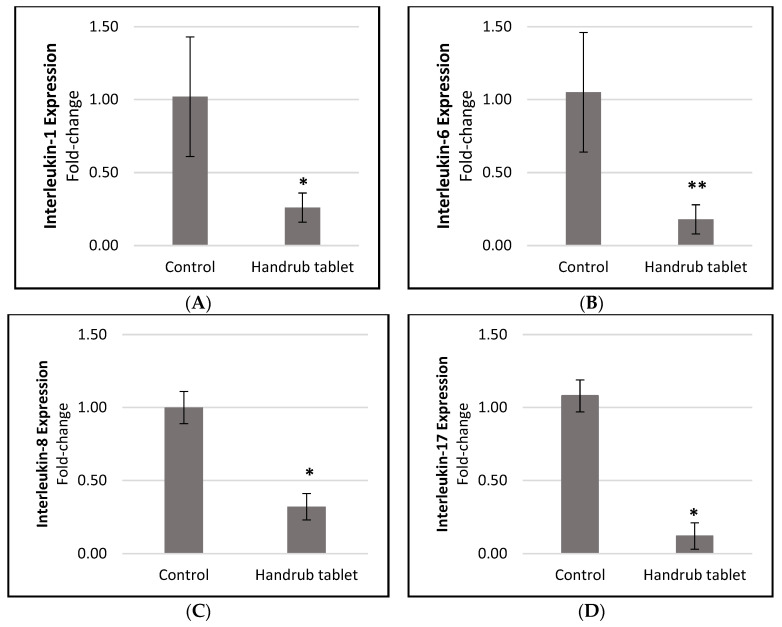
Analysis of the relative expression of (**A**) interleukin-1, (**B**) interleukin-6; (**C**) interleukin-8; (**D**) interleukin-17 for the control and handrub tablet in equivalent skin. * *t*-student analysis with *p* < 0.01; ** *t*-student analysis with *p* < 0.05.

**Figure 10 pharmaceutics-15-02793-f010:**
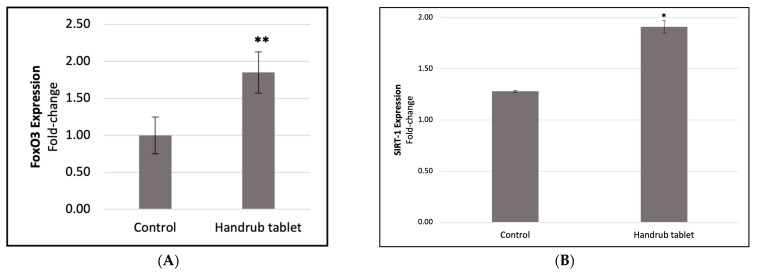
Analysis of the relative expression of (**A**) FoxO3 and (**B**) SIRT-1 for the control and handrub tablet in monolayer cells. * *t*-student analysis with *p* < 0.01; ** *t*-student analysis with *p* < 0.05.

**Figure 11 pharmaceutics-15-02793-f011:**
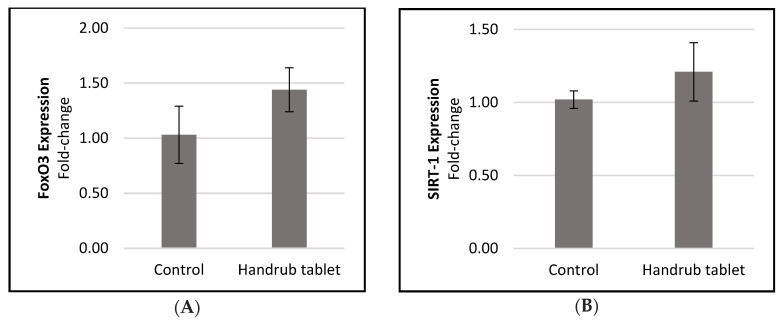
Relative expression of (**A**) FoxO3 and (**B**) SIRT-1 senescence markers following treatment with control and the handrub tablet in the human skin equivalent model.

**Figure 12 pharmaceutics-15-02793-f012:**
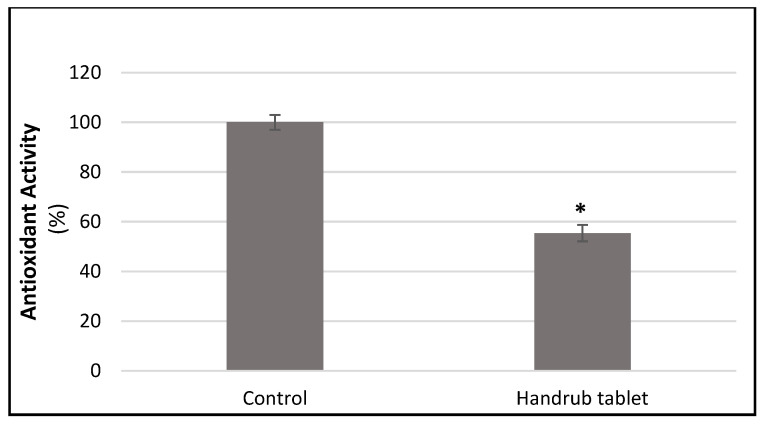
Analysis of the relative decrease in free radical production for the control and handrub tablet in monolayer cells. * *t*-student analysis with *p* < 0.01.

**Figure 13 pharmaceutics-15-02793-f013:**
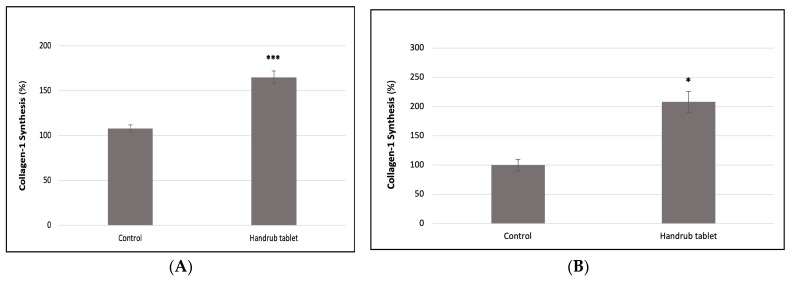
Type 1 Collagen synthesis in human fibroblast cell monolayers (**A**) and in human skin equivalent model (**B**), following control and handrub tablet treatment. *** *t*-student analysis with *p* < 0.001; * *p* < 0.05.

**Figure 14 pharmaceutics-15-02793-f014:**
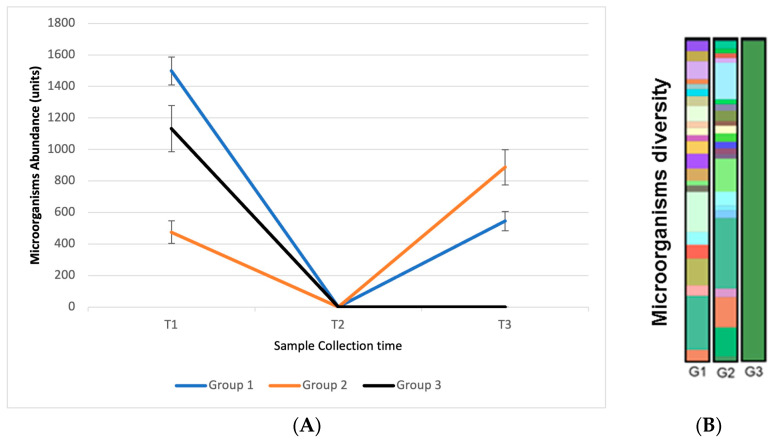
(**A**) Results of the abundance of microorganisms observed among the three groups of participants and the collection times T1, T2, and T3; (**B**) Graphic representation of the diversity of microorganisms observed at collection time T3 between the three groups. Different colors represent microorganisms of different species.

**Figure 15 pharmaceutics-15-02793-f015:**
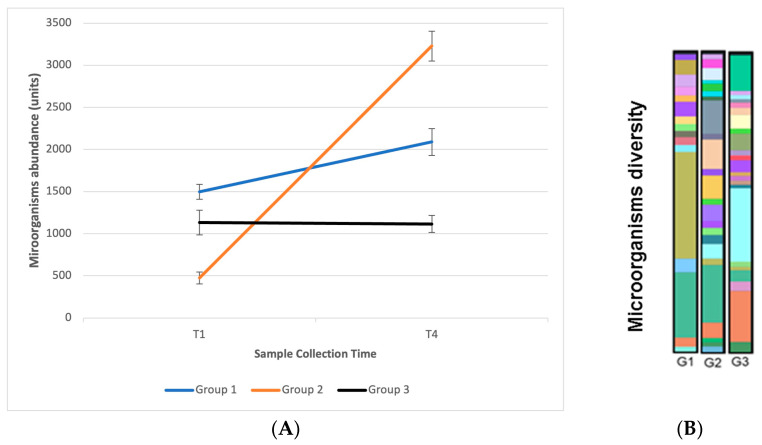
(**A**) The abundance of microorganisms observed among the three groups of participants between collection times T1 (day 1) and T4 (day 8, after 7 days use); (**B**) Graphic representation of the diversity of microorganisms observed at collection time T4 between the three groups. Different colors represent microorganisms of different species.

**Figure 16 pharmaceutics-15-02793-f016:**
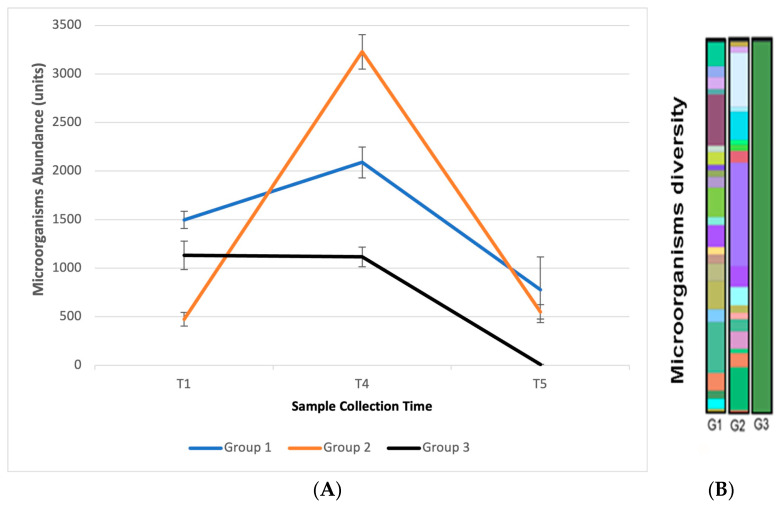
(**A**) Abundance of microorganisms observed among the three groups of participants and collection times T1, T4, and T5; (**B**) Graphic representation of the diversity of microorganisms observed at collection time T5 between the three groups. Different colors represent microorganisms of different species.

**Figure 17 pharmaceutics-15-02793-f017:**
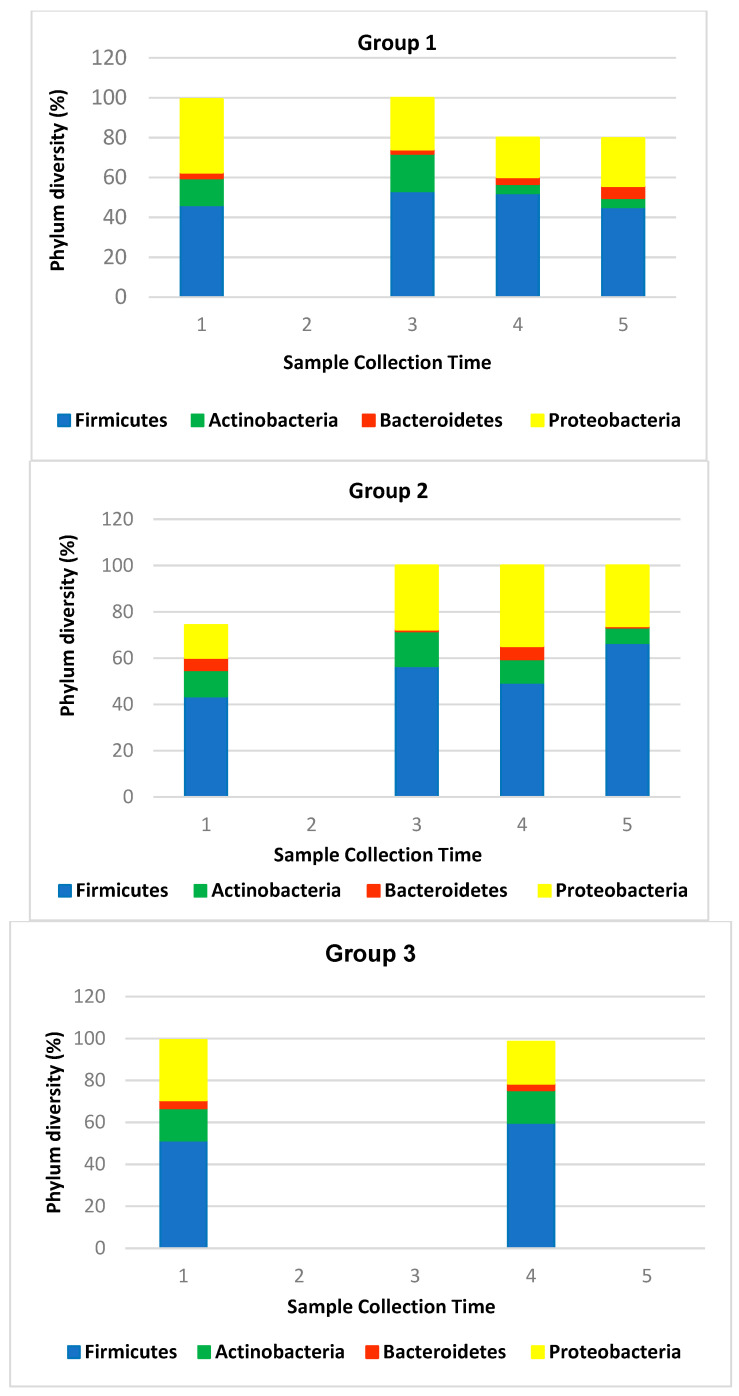
Comparison between the group’s phylum, showing the prevalence of Firmicutes (52.84%) in blue color, in comparison to Actinobacteria (11.50%) in green color, Bacteroidetes (3.12%) in orange color, and Proteobacteria (25.82%) in yellow color.

**Table 1 pharmaceutics-15-02793-t001:** Formulations for base tablet proposed in DoE 2^3^ planning added of a central point.

Components	Percentual Composition (% *w*/*w*)										
	F1	F2	F3	F4	F5	F6	F7	F8	F9	F10	F11
Magnesium stearate	5	5	5	5	5	5	5	5	5	5	5
Colloidal silicon dioxide	1	1	1	1	1	1	1	1	1	1	1
Sodium starch glycolate	10	10	20	20	---	---	---	---	7.5	7.5	7.5
Sodium croscarmellose	---	---	---	---	10	10	20	20	7.5	7.5	7.5
Microcrystalline cellulose	---	84	---	74	---	84	---	74	39.5	39.5	39.5
Pregelatinized Starch	84	---	74	---	84	---	74	---	39.5	39.5	39.5

**Table 2 pharmaceutics-15-02793-t002:** Concentration of ingredients of SLNs.

Ingredients	Concentration (% *w*/*w*)
Cetyltrimethylammonium chloride	1.0
Polyvinylpyrrolidone K30	1.0
Poloxamer 188	1.0
Purified water	90.0
Glyceryl monostearate	1.5
Stearic acid	0.9
Palmitic acid	0.9
Oleic acid	0.6
Lauric acid	0.3
Caprylic acid	1.0
Valeric acid	0.6
Butyric acid	0.2

**Table 3 pharmaceutics-15-02793-t003:** Concentration of handrub tablet.

Ingredients	Concentration (% *w*/*w*)
Blend of Solid lipid nanoparticles and MCC (9% *w*/*w*)	33.0
Prebiotics	16.51
Postbiotics	9.0
Glycyrrhizinate dipotassium	0.15
Menthol	1.0
Fragrance	1.0
Sodium croscarmellose	10.0
Silicon dioxide	1.0
Magnesium stearate	5.0
Hydroxypropyl cellulose	2.0
Microcrystalline cellulose PH101	21.34

**Table 4 pharmaceutics-15-02793-t004:** Properties of tablet base formulations from DoE 2^3^ planning with a central point in triplicate.

Parameters (Mean ± SD) *	Formulations										
	F1	F2	F3	F4	F5	F6	F7	F8	F9	F10	F11
Weight (mg)	89.1	92.9	86.1	91.0	88.7	93.5	87.8	95.6	91.2	89.9	90.0
	± 2.6	± 1.8	± 2.9	± 3.2	± 1.5	± 1.7	± 1.5	± 9.0	± 1.2	± 3.3	± 0.8
Height (mm)	2.03	2.48	2.04	2.44	2.15	2.54	2.11	2.45	2.27	2.24	2.24
	± 0.07	± 0.01	± 0.00	± 0.03	± 0.10	± 0.34	± 0.04	± 0.05	± 0.03	± 0.12	± 0.08
Diameter (mm)	7.89	7.90	7.81	7.82	7.68	7.96	7.73	7.91	7.52	7.88	7.91
	± 0.08	± 0.14	± 0.06	± 0.03	± 0.13	± 0.01	± 0.01	± 0.02	± 0.23	± 0.01	± 0.03
Disintegration time (s)	85.67	22.00	81.00	33.00	51.33	16.00	52.00	26.00	52.67	51.67	56.00
	± 10.02	± 5.57	± 16.09	± 4.36	± 7.77	± 2.65	± 9.17	± 4.36	± 8.39	± 3.21	± 3.00
Swelling (%)	466.9	677.3	610.1	931.1	369.8	421.1	493.9	499.5	656.6	684.9	688.7
	± 54.06	± 19.77	± 30.42	± 54.63	± 3.25	± 11.36	± 38.60	± 24.58	± 2.02	± 13.37	± 7.76
Hardness (N)	<1	<1	<1	2.60 **	2.30 **	6.53	<1	1.63 **	<1	<1	<1
	---	---	---	---	---	1.10	---	---	---	---	---

* mean ± standard deviation; ** only one or two readings SD = standard deviation.

**Table 5 pharmaceutics-15-02793-t005:** Size analysis of SLNs.

Formulation/Analysis	Size Values (µm) (Mean ± SD)				Dispersity (Spam)	Uniformity Ratio
	d10	d50	d90	dm		
Emulsion/passing volume *	0.21 ± 0.03	2.91 ± 0.49	9.80 ± 0.69	3.99 ± 0.40	3.33 ± 0.33	41.27 ± 2.46
Solid lipid nanoparticles dispersion/passing volume **	0.06 ± 0.00	0.22 ± 0.01	0.59 ± 0.04	0.31 ± 0.02	2.43 ± 0.13	6.27 ± 0.42
Solid lipid nanoparticles dispersion/passing number **	0.02 ± 0.00	0.03 ± 0.00	0.07 ± 0.01	0.05 ± 0.00	1.78 ± 0.19	---

* Emulsion before high pressure homogenization; ** Emulsion after high pressure homogenization (results expressed in number and passing volume) SD = standard deviation.

**Table 6 pharmaceutics-15-02793-t006:** Thermal analysis (TGA/DTG and DSC) results of liquid fatty acids raw material, oil phase, and lipid nanoparticles.

Samples	Parameters Recorded			
	Δw (%) (T Range/°C)	Tonset (°C)	Tpeak DTG (°C)	Tpeak DSC (°C)
Butyric Acid	99 (36–173)	151	159	160
Valeric Acid	104 (44–188)	166	179	179
Caprylic Acid	91 (52–242)	215	231	232
Oleic Acid	100 (134–490)	287	325 451	101 331 492
Oil Phase non-lyophilized	100 (97–480)	181 403	214 309 425	46 97 212 317 437
Oil Phase lyophilized	95 (94–496)	247 404	309 427	49 105 309 438
Lipid Nanoparticles dispersion	98 (35–474)	98 221 381	104 244 404	108 237 448
Lipid Nanoparticles lyophilized	99 (48–519)	238 390	241 298 409	86 467

T = temperature; Δw = weight loss; T_onset_ = Onset temperature.

**Table 7 pharmaceutics-15-02793-t007:** Size analysis of freeze-dried samples consisting of SLNs and powdered diluents.

Formulation/Analysis	Size Values (µm) (Mean ± SD)				Dispersity (Spam)	Residual Humidity (%)
	d10	d50	d90	dm		
Solid lipid nanoparticles freeze dried (1:1) *	0.08 ± 0.01	0.28 ± 0.05	0.96 ± 0.03	0.41 ± 0.04	3.19 ± 0.47	2.51 ± 0.30

* Values obtained by laser diffraction from supernatant samples after release assays. SD = standard deviation.

**Table 8 pharmaceutics-15-02793-t008:** Physical characterization of the handrub tablet, based on base tablet F6.

Parameters	Handrub Tablet (Mean ± SD) *
Mean weight (mg)	102.6 ± 2.7
Thickness (mm)	2.78 ± 0.05
Diameter (mm)	8.02 ± 0.02
Disintegration time (s)	22.30 ± 2.90
Swelling rate (%)	332.5 ± 12.2
Hardness (N)	<1 ± 0.00

* mean ± standard deviations.

**Table 9 pharmaceutics-15-02793-t009:** Quantification of viral particles in logarithm and correlation with percentage reduction in respect of monolayer cells and human equivalent skin.

Group	Cell Culture	Time (min)	Log Quantification	Reduction Percentage
Negative control	Monolayer and human equivalent skin	1, 5 and 120	Not identified	-
Viral control	Monolayer	1 and 120	10^5.0^	0
	Human equivalent skin	5	10^4.0^	0
Handrub tablet	Human equivalent Skin	5	10^0.5^	99.96%
	Monolayer	1	10^1.0^	99.99%
	Monolayer	120	10^2.0^	99.9%
70% Alcohol	Human equivalent skin	5	10^1.0^	99.9%

## Data Availability

Data are contained within the article and Appendix A.

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
