# Peer review of "A Novel Handrub Tablet Loaded with Pre- and Post-Biotic Solid Lipid Nanoparticles Combining Virucidal Activity and Maintenance of the Skin Barrier and Microbiome"

_pharmaceutics, 2023, doi:10.3390/pharmaceutics15122793_

Round 1
Reviewer 1 Report
Comments and Suggestions for Authors 1) The introduction mostly deals with generalities such as the negative effects of alcoholic sanitizers, the microorganisms found on hands, water conservation, etc. What is needed are some references dealing with the specific subject matter of the study, for example, solvent-free nanocapsules in J Steelandt et al., Antimicrobial nanocapsules: from new solvent-free process to in vitro efficiency, International Journal of Nanomedicine 9 (2014) 4467-4469 and cetyltrimethylammonium/silica in Rie Hirao et al., Virus Inactivation Based on Optimal Surfactant Reservoir of Mesoporous Silica, Applied Bio Materials 6 (2023) 1032-1040. 2) It is not clear how the references on line 98 support the inclusion of DPG, which is misspelled on 97 and 849, in the tablet. Neither reference addresses skin barrier function, 22 is specific for lung inflammation with no benefit of lowering infection risk and 23 describes its therapeutic action in skin wound healing. 3) The intended application of the tablet described on line 338 indicates that the components will start drying out after the initial 90mg tablet/1g water rub. This condition is quite different from the lengthy periods in cell culture medium with constant water content used in the tests in section 2.5, and a comment should be added about how this may affect conclusions from the study. The 4) ASTM-D5806-95-2017 is designed to determine the active materials in disinfectant products based on quaternary ammonium salts, and a comment should be added as to how the handrub tablet would perform in such a test. 5) Solutions are mentioned a number of times in section 2.5 in connection with additions to cell cultures. The word is used again in Suppl. 4 for MCC concentrations with no accompanying information in the text. It seems illogical that solution 1 which was anhydrous had the lowest swelling rate since the other samples already had water included in their starting weights. 6) DoE was employed to arrive at the optimal F6 tablet base formulation according to section 2.2. In contrast to this methodical procedure, other compositions were only described as proposed ranges of concentrations. The solid lipid particles were composed of the ingredient variations shown in Table 2, the final handrub tablet with the ingredient variations shown in Table 3 with its physical characteristics given in Table 8. There thus doesn't seem to be any exact composition given for the final handrub tablet used for all the safety and efficacy tests. 7) The term nano is usually used to designate particles in the size range <100 nm. What is described in the manuscript is thus lipid microparticles. While this is only a semantic nicety, of more practical interest is the statement on 224 about the ability of the SLNs to enter the aqueous medium. This raises the question if a size distribution more towards the nanometer range would improve the desorption properties, especially considering that no optimization of the SLN particle size was attempted?Author Response
Please see the attachment

Reviewer 2 Report
Comments and Suggestions for Authors
The author developed the novel handrub tablet loaded with pre- and post-biotic solid lipid nanoparticles well and good.
The solid lipid nanoparticles are not properly characterized. Should be characterized XRD, FTIR, TEM, EDAX and other related studies.
Give me a proper mechanism for interaction of solid lipid NPs with microbes.
Should be included more microbial related characteristics like SEM for morphology analysis, MIC, and MBC.
Should be included synthesis scheme.
Comments on the Quality of English LanguageThe author developed the novel handrub tablet loaded with pre- and post-biotic solid lipid nanoparticles well and good.
The solid lipid nanoparticles are not properly characterized. Should be characterized XRD, FTIR, TEM, EDAX and other related studies.
Give me a proper mechanism for interaction of solid lipid NPs with microbes.
Should be included more microbial related characteristics like SEM for morphology analysis, MIC, and MBC.
Should be included synthesis scheme.
